# CooperTrim: Adaptive Data Selection for Uncertainty-Aware Cooperative Perception

**Shilpa Mukhopadhyay**[1,2][*]**, Amit Roy-Chowdhury**[1]**, Hang Qiu**[1]

[1]University of California, Riverside, USA. [2]New Jersey Institute of Technology, New Jersey, USA.
{sm3934}@njit.edu, {amitrc, hangq}@ucr.edu

## Abstract

Cooperative perception enables autonomous agents to share encoded representations over wireless communication to enhance each other's live situational awareness. However, the tension between the limited communication bandwidth and the rich sensor information hinders its practical deployment. Recent studies have explored selection strategies that share only a subset of features per frame while striving to keep the performance on par. Nevertheless, the bandwidth requirement still stresses current wireless technologies. To fundamentally ease the tension, we take a proactive approach, exploiting the temporal continuity to identify features that capture environment dynamics, while avoiding repetitive and redundant transmission of static information. By incorporating temporal awareness, agents are empowered to dynamically adapt the sharing quantity according to environment complexity. We instantiate this intuition into an adaptive selection framework, CooperTrim, which introduces a novel conformal temporal uncertainty metric to gauge feature relevance, and a data-driven mechanism to dynamically determine the sharing quantity. To evaluate CooperTrim, we take semantic segmentation and 3D detection as example tasks. Across multiple open-source cooperative segmentation and detection models, CooperTrim achieves up to 80.28% and 72.52% bandwidth reduction respectively while maintaining a comparable accuracy. Relative to other selection strategies, CooperTrim also improves IoU by as much as 45.54% with up to 72% less bandwidth. Combined with compression strategies, CooperTrim can further reduce bandwidth usage to as low as 1.46% without compromising IoU performance. Qualitative results show CooperTrim gracefully adapts to environmental dynamics, localization error, and communication latency, demonstrating flexibility and paving the way for real-world deployment. Project website link for code and pretrained models: https://cisl.ucr.edu/CooperTrim.

## 1 Introduction

Cooperative perception (Wang et al., 2020; Xu et al., 2023a) enables autonomous agents to exchange information about invisible or uncertain regions to enhance a range of tasks, including detection (Qiu et al., 2022), prediction (Wang et al., 2025), mapping (Ahmad et al., 2020), and navigation (Cui et al., 2022; 2026). The benefits of additional vantage points come at the cost of communication. Balancing bandwidth overhead and accuracy, modern approaches adopt intermediate (over early and late) fusion (Wang et al., 2020; Xu et al., 2023a) to share task-oriented lightweight encoded representations of the environment. To further improve communication efficiency in intermediate fusion, three strategies have been proposed: *compression* (Wang et al., 2020; Xu et al., 2022a), which reduces the size of the features but risks information loss if targeted at higher compression ratio (*i.e.,* lossy compression); *selection* (Hu et al., 2023; Liu et al., 2020), which transmits only useful data or selects agents; and *hybrid* (Yuan et al., 2022; Yang et al., 2023), combining both for optimal bandwidth reduction. For example, Where2comm (Hu et al., 2022) uses threshold-based spatial confidence maps for feature selection, UniSense (Ren et al., 2025) focuses on uncertainty-driven data exchange, and SwissCheese (Xie et al., 2024) exploits the disparity in semantic information

---

[*]Work done while affiliated with the University of California, Riverside.

on features between different spatial regions on different channels to perform fixed threshold-based selection. While these approaches reduce the exchange volume, the demand still strains wireless bandwidth (Mo et al., 2025).

Fundamentally, the mismatch between limited communication bandwidth and the richness of sensor information hinders the practical deployment of cooperative perception. To address this mismatch, we take a proactive adaptation approach. Our intuition is twofold: 1) rather than sharing a smaller yet static volume of features per frame, sharing should be made on demand at a variable volume depending on the recipient's cognitive challenges for its surrounding environment; 2) rather than sharing frame by frame independently, cooperation should leverage the temporal context just as the rest of the autonomy stack does to enhance temporal comprehension while reducing repetitive and redundant sharing. Consider a scenario where the ego (recipient) agent is confident in most features from its sensor data, but collaborating agents may have varying confidence levels of elements in the scene. In such cases, the ego can learn from its temporal context to request only the uncertain features. For instance, a complex scenario (*e.g.,*multiple road intersections) should demand more features over consecutive frames than a simple scenario (*e.g.,*no intersections).

Building on top of these insights, in this paper, we present COOPERTRIM, an adaptive cooperative perception framework, which introduces a novel conformal temporal uncertainty metric to gauge feature relevance, and a data-driven mechanism to dynamically determine the sharing quantity. Our methodology employs temporal uncertainty estimation using conformal prediction inspired quantile gating mechanism to identify feature deviations across frames, alongside an uncertainty-based attention mechanism to weigh feature importance. It dynamically selects relevant features with adaptive thresholds based on environmental needs and facilitates efficient feature exchange between agents to minimize bandwidth usage while ensuring robust fusion of multi-source data.

We instantiate the framework using semantic segmentation as an example cooperative task, and evaluate it against existing open-source cooperative segmentation models (Xu et al., 2023a; 2022b; Li et al., 2021) and existing selection strategies. To our knowledge, this is the first work on selective cooperative semantic segmentation. Unlike detection or tracking, which can work with sparse, object-level features, segmentation demands pixel-level granularity for exact shapes and boundaries, which poses more challenges for bandwidth reduction—a difficult case to address the bandwidth-information mismatch. We evaluate COOPERTRIM through extensive benchmarking on the OPV2V (Xu et al., 2022b) and V2V4Real (Xu et al., 2023b) datasets, assessing the performance and network overhead across multiple open-source cooperative 3D detection and segmentation models, as well as against other existing selection strategies. COOPERTRIM performs comparable to baselines with a bandwidth reduction of 80.28% (for segmentation) and 72.52% (for 3D detection) on average. Furthermore, we perform a wider network overhead analysis by comparing the bandwidth consumption against 10 existing baselines. COOPERTRIM demonstrates competitive IoU performance compared to baselines for compression in controlled settings, using only 1.46% bandwidth at 32x compression. To understand the efficacy of the adaptation design, we evaluate data request variations across frames with respect to environment complexity. Our evaluation also shows robustness to localization error and communication latency.

In summary, COOPERTRIM makes the following contributions:

- We propose a learning framework to proactively adapt feature selection by dynamically determining feature *relevance* and sharing *quantity*—a transformative deviation from existing static selection frameworks.
- We propose a novel relevance assessment strategy by quantifying temporal uncertainty using conformal prediction inspired quantile gating meachanism to compare features across frames. Coupled with the conformal assessment using an adaptive quantile threshold based on conformity score, we use an attention mechanism with an adaptive mask threshold for quantity estimation.
- Our instantiation of COOPERTRIM on cooperative segmentation is the first work that demonstrates feature selection on this task. Segmentation requires transmitting large volumes of data, making selective perception challenging under bandwidth constraints. Our approach addresses this by intelligently selecting and prioritizing critical features, reducing data transmission while preserving segmentation accuracy.
- We employ a training method inspired by the $\epsilon$-Greedy exploration strategy from reinforcement learning (Liu et al., 2022), which balances exploration and exploitation effectively in training, resulting in lower bandwidth and higher task performance.

- Our evaluation shows that COOPERTRIM achieves up to 80.28% and 72.52% bandwidth reduction in cooperative segmentation and detection respectively while maintaining a comparable accuracy. Relative to other selection strategies, COOPERTRIM also improves IoU by as much as 45.54% with up to 72% less bandwidth. Quantitative frame-by-frame inspection further validates the flexibility, demonstrating graceful adaptation to environmental dynamics and paving the way towards real-world deployment.

## 2  RELATED WORK

**Cooperative Perception.** Resilient operation of autonomous agents depends on their onboard perception, which is often limited by blind spots and uncertainties. A range of cooperative perception mechanisms have been proposed, from early fusion and edge assistance (Zhang et al., 2021; Zhu et al., 2024) sharing raw sensor data, to late fusion of regional processing results while missing holistic scene details. The most prominent are intermediate feature fusion methods (Chen et al., 2019), however, there remains a gap between bandwidth demand and availability. V2VNet (Wang et al., 2020) uses compressed LiDAR BEV features and GNN aggregation, yet often transmitting redundant data. AttFuse (Xu et al., 2022b) and DiscoNet (Li et al., 2021) offer attentive fusion and collaboration graphs, but lack intelligent data selection. CoBEVT (Xu et al., 2023a) integrates multi-vehicle BEV features via SinBEVT and FuseBEVT for segmentation, yet struggles with inefficient feature compression. Unlike these approaches, COOPERTRIM selectively transmits essential perception features, reducing bandwidth usage, improving network efficiency, and maintaining accuracy under constraints.

**Network Efficient Cooperation.** The literature on communication-efficient cooperative perception includes Compression-based, Selection-based, and Hybrid approaches. We focus on Selection-based methods due to practical wireless broadcasting needs. Where2comm (Hu et al., 2022) uses threshold based spatial confidence maps for selection, ignoring uncertainties and relying on impractical multi-round transmission. UMC (Wang et al., 2023) applies entropy-based selection but sending full maps is computationally expensive. CenterCOOP (Zhou et al., 2023) transmits center-point LiDAR features via mutual information, improving bandwidth by 10% on DAIR-V2X while sharing all embeddings. BM2CP (Zhao et al., 2023) shares modality-guided features, overlooking sensor noise. UniSense (Ren et al., 2025) selects critical regions via uncertainty, missing occlusion uncertainties and incurring high bandwidth costs. SwissCheese (Xie et al., 2024) uses fixed thresholds for selection, lacking adaptability to dynamic scenarios. Different from methods with fixed thresholds or static selection strategy, COOPERTRIM adapts feature selection to environment dynamics, prioritizes uncertainty-driven critical features using past confident data, balancing bandwidth efficiency and perception performance resilience.

## 3  METHODOLOGY AND SOLUTION

### 3.1  PROBLEM STATEMENT

We address the challenge of selective feature sharing for cooperative perception tasks. Rather than designing a new model, we aim at a selection framework that is applicable to a set of similar feature-sharing mechanisms (Xu et al. (2023a); Hu et al. (2022); Li et al. (2021). Given a set of features $\mathcal{F} = \{f_1, f_2, \ldots, f_n\}$ extracted from input data, where $n$ is the number of feature channels in latent representations, our objective is to identify a subset $\mathcal{S} \subseteq \mathcal{F}$ such that the number of selected features $|\mathcal{S}|$ is minimized, while the accuracy of the perception task, denoted as a function $A(\mathcal{S})$ of the selected features, is maximized. Formally, the optimization problem can be expressed as $\min_{\mathcal{S} \subseteq \mathcal{F}} |\mathcal{S}|, \max_{\mathcal{S} \subseteq \mathcal{F}} A(\mathcal{S})$. This bi-objective optimization problem is challenging due to the *inherent **tension** between feature abundance and bandwidth limitations*.

### 3.2  COOPERTRIM DESIGN

To resolve that tension, we present COOPERTRIM, a selective cooperative perception framework that learns to enhance representation learning via temporal uncertainty-driven feature selection for bandwidth-efficient, accurate perception in multi-agent systems. COOPERTRIM addresses two key research questions:

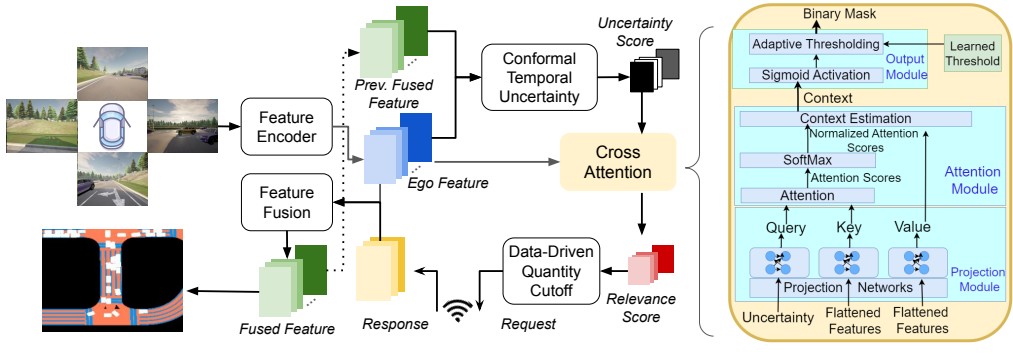

(a) COOPERTRIM Overview       (b) Cross-Attention Module

Figure 1: (a) COOPERTRIM Overview. COOPERTRIM conducts feature learning, followed by an uncertainty-based selection module using learned features. It estimates adaptive temporal uncertainty (via learned confidence) for each feature, performs cross-attention-based feature weighting, and selects features using a learned threshold. The ego then broadcasts a request vector for selected features, reconstructs received CAV data into full features, fuses them, and sends them to the task head for final results. (b) Cross-Attention Module uses learned projections of temporal uncertainty as queries, and feature projections as keys and values. These matrices pass through an attention module, and a learned threshold at the final output generates a binary mask for selected channels.

- **Relevance.** If the bandwidth does not permit all features to be shared, what are the most essential features that are impactful to downstream tasks? The relevance is inevitably recipient-centric, which demands the recipient vehicle to assess its field-of-view visibility, perception uncertainty, and environment dynamics.
- **Quantity.** If most relevant features are prioritized, where is the diminishing return point to stop sharing? Such quantity sweet spot may be dependent on both scene- and task-complexity, which demands dynamic runtime adaptation, a sophistication in COOPERTRIM framework design.

COOPERTRIM leverages two key insights in addressing the above research questions. i) *Temporal Uncertainty as Relevance.* Unlike conventional approaches Ren et al. (2025); Xie et al. (2024); Hu et al. (2022), which quantify the feature relevance of each individual frame, we put the features in their temporal contexts, and quantify the temporal uncertainty (*e.g.,* introduced by scene dynamics, changing lighting conditions, occlusions) as relevance to determine sharing priority. We use conformal prediction inspired quantile gating method to better assess the temporal uncertainty in continuous frames. ii) *Environment Adaptivity.* Adapting to environment dynamics, we introduce two learned parameters: a data-driven uncertainty threshold used in the uncertainty estimation, and a data-driven attention mask threshold, to adjust the runtime sharing quantity for each frame. This flexibility empowers COOPERTRIM to scale gracefully while maintaining efficient bandwidth utilization. Figure 1 shows an overview of COOPERTRIM. We describe the components in detail.

**Conformal Temporal Uncertainty.** COOPERTRIM takes encoded representations from sensor data, contextualizes them within the ego agent's recent memory, and identifies the uncertain ones as candidates for enhancement through cooperation. Specifically, given current frame feature $F_t \in \mathbb{R}^{H' \times W' \times D}$, encoded from sensor input $X_t \in \mathbb{R}^{H \times W \times C}$, it calculates conformity scores by comparing the feature $F_t$ against the fused features from previous frames, denoted as distance function $S_t = d(F_t, F_{t-1}^{\text{fused}})$, where $d(.)$ is the L1 distance defined as $|F_t - F_{t-1}^{\text{fused}}|$, effectively capturing deviations between past and current scene understanding. In this context, the use of 'conformal' is inspired by conformal prediction, differing by (a) learning online frame-wise instead of using a fixed calibration dataset, and (b) estimating uncertainties in components of regression values rather than direct intervals on entire regression data. Then, inspired by conformal prediction, we use quantile gating to assess the features' relevance, introducing a learnable quantile threshold $q$, and apply a cross-attention mechanism between the features and the ranges of those scores above the threshold (*i.e.,* $S_t(f) > q$) to obtain the relevance metric $R_t$.

**Data-driven Quantity Cutoff.** In order to adapt to environment dynamics, we introduce a mechanism to determine the diminishing return point of the sharing volume. Specifically, we introduce a second learnable threshold $\tau$ such that we only share those features whose relevance score is above

the threshold ($R_t > \tau$). In situations where the perception complexity is high, the range of conformity score will increase due to more temporally diverse feature encodings, which, in turn, yields higher relevance scores. More features would have relevance scores above the learned threshold; the communication of which incurs higher bandwidth usage. On the contrary, temporal consistency will result in low relevance scores among all features, which yields low sharing volume. Moreover, the temporal consistency not only comes from the low environment dynamics, but also comes from perception stability—COOPERTRIM leverages this stability to save bandwidth as well.

**Feature Exchange and Fusion.** To communicate recipient-centric relevance assessment, the ego agents send requests for selected high-relevance features. Following prior work assumptions of precise pose estimation (Xu et al., 2023a), responders use spatial transformation (Jaderberg et al., 2015) to match ego's perspective and share the requested features. The ego agent blends received features into its own feature map before passing them along to the fusion decoders and task heads.

## 3.3  TRAINING

**Loss Function.** To attain the joint goal of bandwidth efficient transmission and task performance, the training of COOPERTRIM can be formulated as a constrained optimization problem as follows:

$$\theta^* = \arg\min_{\theta} L(C(\theta)), \text{s.t. } P(C(\theta)) = C_{1.6} \tag{1}$$

where $C_{1.6}$ represents the percentage of channels corresponding to 1.6 Mbps (Mo et al., 2025), $C(\theta)$ represents the channels selected, $P(\cdot)$ represents the precentage of channels selected, $L(\cdot)$ represents the perception task loss. To solve this, we use the Lagrangian formulation, defining the total loss as

$$\theta^* = \arg\min_{\theta} L(C(\theta)) + \lambda \cdot (P(C(\theta)) - C_{1.6}) \tag{2}$$

where $\lambda$ acts as a Lagrange multiplier dynamically adjusted (explained in Appendix A.1) to enforce the constraint over time. The strategy starts with unconstrained optimization for initial learning, then introduces and intensifies constraint enforcement over time, using periodic strong adjustments for major deviations and steady increments for fine-tuning.

**Training Strategy.** For training the loss function in COOPERTRIM, we use an $\epsilon$-Greedy method inspired by the reinforcement learning $\epsilon$-greedy exploration strategy (Liu et al., 2022). However, we make some adaptations for our training purpose. With $\epsilon$ probability we request entire feature set and with $(1-\epsilon)$ probability we exploit the knowledge. By exploiting the knowledge, we request the data as per learned thresholds (i.e. conformal predictions's quantile threshold $q$ and cross-attention module's mask threshold $\tau$). We see considerable improvement in bandwidth requirement with this training method. Occasional updates considering all features can stabilize the optimization trajectory by smoothing out erratic updates caused by noise from partial features. This can prevent the model from diverging or oscillating around suboptimal points, leading to smoother convergence to a better solution. The $\epsilon$-Greedy method balances exploration (full data $D_{\text{full}}$ with probability $\epsilon$) and exploitation (partial data $D_{\text{partial}}$ with probability $(1 - \epsilon)$). This balance stabilizes optimization by reducing gradient noise through periodic full-data updates. As formalized in Proposition 1, this strategy reduces both the bias and variance of the gradient estimator compared to using only partial data. The expected gradient

$$\mathbb{E}[\nabla L] = \epsilon \cdot \mathbb{E}[\nabla L(D_{\text{full}})] + (1 - \epsilon) \cdot \mathbb{E}[\nabla L(D_{\text{partial}})] \tag{3}$$

ensures smoother convergence while adhering to bandwidth constraints. We formalize the effectiveness of the $\epsilon$-Greedy training strategy in the following proposition.

**Theoretical Analysis.** We draw a theoretical analysis based on the following assumptions. (1) Perfect transmission conditions (no noise introduced during transmission, so any bias or variance arises solely from the data used), (2) A complete dataset $D_{\text{full}}$ and a subset $D_{\text{partial}} \subseteq D_{\text{full}}$ representing partial data, (3) The true gradient is $\nabla L(D_{\text{full}})$, reflecting the loss over the entire dataset.

**Proposition 1** (Effectiveness of $\epsilon$-Greedy Training)**.** *An $\epsilon$-greedy training strategy that computes the gradient of Loss L using full data ($D_{full}$) with probability $\epsilon$ and partial data ($D_{partial}$) with probability $(1 - \epsilon)$ reduces the bias of the gradient estimator compared to using only partial data. Specifically, the bias is scaled down by a factor of $(1 - \epsilon)$.*

*Proof.* Detailed proof provided in Appendix A.2. □

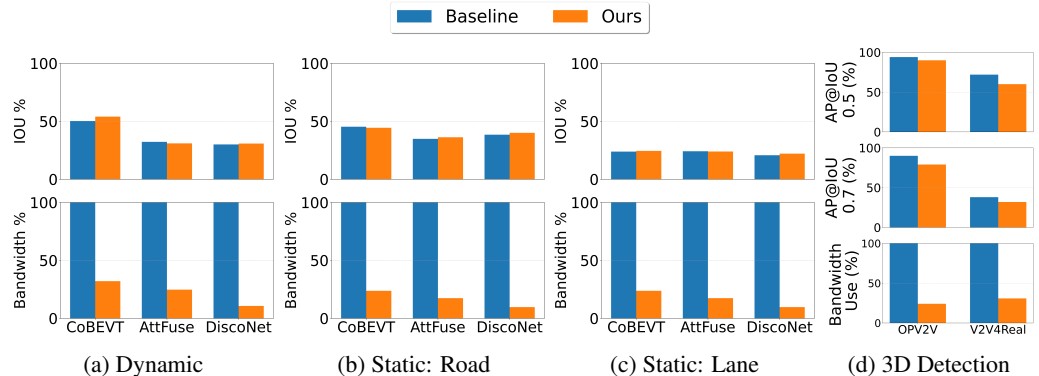

Figure 2: "Trimming" existing cooperative perception baselines, COOPERTRIM reduces bandwidth significantly while preserving accuracy in segmentation (across different semantics, *i.e.,* dynamic, static road, and static lane) and 3D detection tasks.

## 4  EXPERIMENTS

We conduct six experiments to demonstrate COOPERTRIM's efficiency in task performance and bandwidth usage. Referred to as COOPERTRIM, we apply COOPERTRIM to CoBEVT (Xu et al., 2023a) for evaluation, focusing on semantic segmentation on the OPV2V (Xu et al., 2022b) dataset and 3D detection on OPV2V and V2V4Real (Xu et al., 2023b) datasets. We choose CoBEVT for its robust performance in cooperative perception compared to existing works. COOPERTRIM is the first to address selective cooperative perception in semantic segmentation.

**"Trimming" Existing Cooperative Perception Baselines.** In Figure 2, we assess performance and bandwidth (BW) by applying our temporal uncertainty-aware method COOPERTRIM to three methods: CoBEVT (Xu et al., 2023a), Attfuse (Xu et al., 2022b), and DiscoNet (Li et al., 2021). We implement COOPERTRIM on these methods after removing any existing compression techniques. This evaluation compares performance accuracy between COOPERTRIM-based selection and the original methods (without compression/selection), and reports bandwidth usage percentages relative to the originals (40 Mbps). For Dynamic and Static segmentation combined, COOPERTRIM-CoBEVT, COOPERTRIM-Attfuse, and COOPERTRIM-DiscoNet use average bandwidths of 27.9%, 21.07%, and 10.18%, respectively, corresponding to 11.16 Mbps, 8.4 Mbps, and 4.07 Mbps based on our 128x32x32 latent representation size. We observe that COOPERTRIM achieves comparable performance accuracies to the original methods despite significantly lower bandwidth consumption, with an average 80.28% improvement in network overhead over the baselines. Detailed values are in Appendix Table 4.

Additionally, we validate COOPERTRIM's generalizability on 3D detection using LiDAR data on OPV2V (Xu et al., 2022b) and V2V4Real (Xu et al., 2023b) datasets. Figure 2d shows COOPERTRIM's performance versus baseline CoBEVT (Xu et al., 2023a) at IoU 0.5 and 0.7. COOPERTRIM maintains comparable detection accuracy with 27.48% bandwidth usage (72.52% reduction), highlighting its efficiency in lowering network overhead.

**Selection Strategy Comparison.** Table 1 presents the evaluation of COOPERTRIM against selection-based, network-efficient baselines. Our feature latent representation is sized at 128x32x32, with percentages and bandwidths reported accordingly. We implement two algorithms: Where2Comm (Hu et al., 2022) (feature and agent selection, adapted for segmentation with a 0.4 threshold on batch confidence map) and SwissCheese (Xie et al., 2024) (feature selection over spatial and channel features, implemented in our camera-based segmentation framework as the official code is unavailable). For SwissCheese, we set bandwidth at 10 Mbps (comparable to COOPERTRIM, equating to 25% of our latent representation size) to assess segmentation performance. COOPERTRIM achieves lower bandwidth usage and better performance than Where2Comm, and outperforms SwissCheese in dynamic and static accuracy at similar bandwidth levels.

**Post-selection Compression Compatibility.** Compression-based methods like CoBEVT (Xu et al., 2023a) and AttFuse (Xu et al., 2022b) reduce data size for network efficiency. Yet, COOPERTRIM's

Table 1: Feature Selection Strategy Comparison: Accuracy and Bandwidth

| Baselines | Accuracy (IoU, %) | | | Bandwidth (Mbps) |
|---|---|---|---|---|
| | Dynamic | Static Lane | Static Road | |
| Where2Comm (Hu et al., 2022) | 8.62 | 20.40 | 36.46 | 39.6 |
| SwissCheese (Xie et al., 2024) | 35.71 | 12.81 | 32.07 | 10 |
| **COOPERTRIM (ours)** | **54.03** | **24.45** | **44.38** | 11.16 |

data selection enhances efficiency by compressing post-selection, optimizing bandwidth. We apply lossy quantization to reduce precision and volume, then lossless methods for further reduction. Figure 3 compares IoU and bandwidth for COOPERTRIM, CoBEVT (Xu et al., 2023a), and AttFuse (Xu et al., 2022b) at 1x, 8x, and 32x rates for the segmentation task. COOPERTRIM excels in IoU and bandwidth savings, notably at 32x (1.46% vs. 3.76% for CoBEVT and 10.62% for AttFuse in 32x Dyn) [See Appendix A.6 for detailed results].

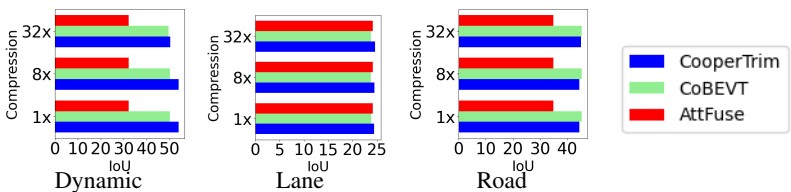

Figure 3: Comparison of IoU performance and bandwidth usage at compression rates (1x, 8x, 32x) for COOPERTRIM, CoBEVT, and AttFuse in Dynamic, Road, and Lane scenarios.

**Environment Adaptation.** Figure 4 shows the inference results for COOPERTRIM over multiple frames. We see variations in bandwidth implying variable feature request in each frame. The figure shows increased data requests correlate with higher scene complexity. In segmenting the dynamic objects, an increase in complexity can be referred to criticality in positioning of an increased amount of vehicles and traffic participants. Similarly, an increase in complexity in the static task can be referred to an increase in the number of intersections or lane orientations in the scene. Figure 4 shows that in both cases, *i.e.,* increased amount of vehicles, and increased topological complexity in the roadways, COOPERTRIM accordingly shares more data to preserve task performance, whereas in other cases, gracefully adapts to request less sharing, saving the precious bandwidth to other agents in need. In critical frame ranges such as 1000-1800 (Dynamic), 1000-1500 (Lane), and 1200-1400 (Road), where baseline CoBEVT (Xu et al., 2023a) consistently underperforms, COOPERTRIM achieves much higher IoU, illustrating the effectiveness of its adaptive threshold masking mechanism in prioritizing key features and managing scene complexity (see Appendix A.5).

**Training Methods Comparison.** Table 2 shows the comparative study for training COOPERTRIM using different training methods, all using the loss function in Section 3.3. **(1)** COOPERTRIM is trained with $\epsilon$-Greedy (EG) based fine-tuning (FT) using Conformal Prediction (CP) for temporal

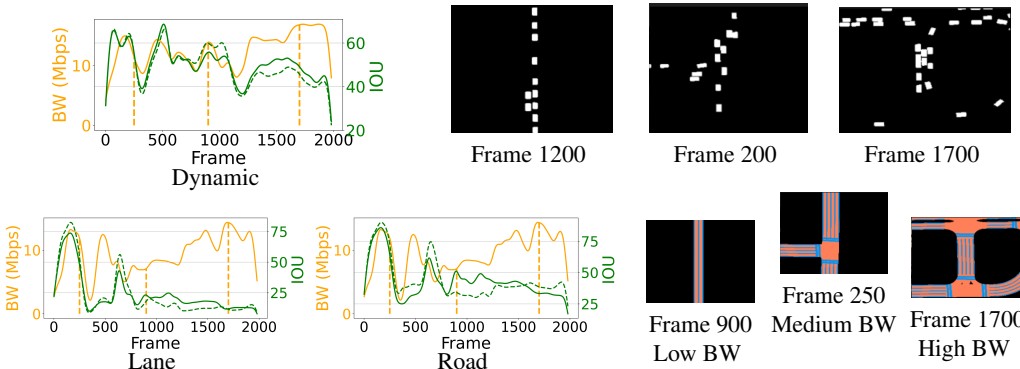

Figure 4: Increased data requests align with higher scene complexity. For dynamic objects, complexity in number and positioning rises in Frames 1200, 200, and 1700. For static elements, complexity grows in Frames 900, 250, and 1600, with more intersections and lane orientations. The right part highlights the frames at vertical lines; Frame 1200, 200, 1700 (dynamic) and 900, 250, 1600 (static). Green dashed lines: baseline CoBEVT IoU. Green solid lines: COOPERTRIM IoU.

Table 2: Training Methods Comparison for Uncertainty Aware Training showing COOPERTRIM (EG+CP+FT) striking a good balance between Accuracy and Bandwidth

| Training Methods | Accuracy (IoU, %) | | | Bandwidth (%) | |
|---|---|---|---|---|---|
| | Dynamic | Static Lane | Static Road | Dynamic | Static |
| CP+FT | 51.57 | 23.05 | 36.92 | **30.75** | **1.12** |
| Curriculum+CP+FT | 52.47 | **28.27** | **45.44** | 49.25 | 51.63 |
| EG+SD+FT | 52.55 | 23.69 | 38.45 | 35.64 | 11.03 |
| EG+CP+FT (**Ours**) | **54.03** | 24.45 | 44.38 | 32.04 | 23.77 |

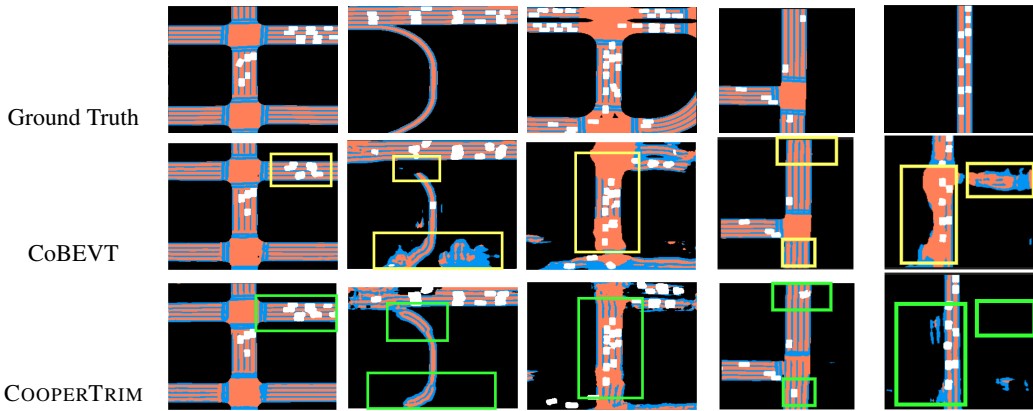

Figure 5: Qualitative Comparison of Ground Truth, CoBEVT, and COOPERTRIM. COOPERTRIM performs better than CoBEVT in estimating dynamic object segments (Column 1 and 4), road segments (Column 2 and 5), and lane segments (Column 3 and 5).

uncertainty estimation [Section 3.3], termed $EG + CP + FT$. **(2)** The first method uses Standard Deviation (SD) for spatial uncertainty estimation over latent feature channels ($EG + SD + FT$), showing higher network consumption for dynamic objects (35.64% vs. COOPERTRIM at 32.04%) due to lack of temporal consideration in SD, and 6% lower performance in static road segmentation. **(3)** The second method employs a basic curriculum for fine-tuning ($Curriculum + CP + FT$) with four stages: (a) basic fine-tuning, (b) cross-attention with fixed mask threshold and CP confidence, (c) adaptive mask threshold with fixed CP confidence, and (d) fully adaptive (learned mask and CP confidence) CP-guided selection. This yields good performance but higher bandwidth use (49% dynamic, 51% static). **(4)** The final method omits EG in FT ($CP + FT$), resulting in lower accuracy (static road 8% below COOPERTRIM). EG's occasional full-feature updates stabilize optimization by promoting smoother convergence.

**Qualitative Results.** Figure 5 shows the segmentation results of COOPERTRIM to analyse visually. COOPERTRIM is different from CoBEVT (Xu et al., 2023a) only in its feature set before fusion of muti-source data. COOPERTRIM performs better than CoBEVT in estimating road segment, dynamic object segment, and lane segment. This is attributed to the better feature representation achieved by COOPERTRIM selection mechanism, which reduces uncertainty in feature, by focusing on selected data transmission rather than transmitting the entire dataset for fusion.

**Sensitivity Analysis.** COOPERTRIM's performance is tested across localization errors (0cm, ±20cm, ±1m) during inference, as shown in Figure 6. COOPERTRIM remains robust to small errors (up to ±20cm) and maintains comparable IoU at ±1m, with stable bandwidth usage, indicating consistent communication efficiency. COOPERTRIM's performance under latency (0ms to 200ms) is shown in

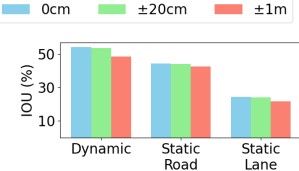
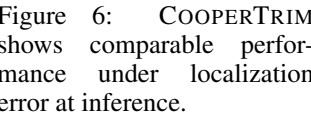

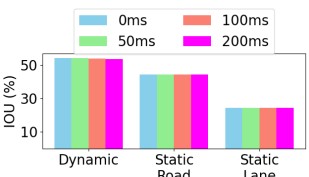

Figure 6: COOPERTRIM shows comparable performance under localization error at inference.

Figure 7: COOPERTRIM shows comparable performance under latency at inference.

Figure 7. It exhibits strong robustness up to 50ms with no loss, and marginal IoU drop at 100ms and 200ms. This resilience stems from COOPERTRIM's adaptive mechanisms, two-threshold policy, and uncertainty-driven requests, reducing reliance on misaligned data from CAVs. By requesting less data (23-32%, see details in Appendix Table 4), latency impact is minimized, enhancing real-world robustness. Network usage stayed consistent during testing.

**Network Overhead Comparison.** We compare the network overhead of COOPERTRIM with used bandwidths of a broader range of existing cooperative perception works. We evaluate Feature Selection (FS) algorithms UniSense (Ren et al., 2025), SwissCheese (Xie et al., 2024), Compression-based (C) algorithms STAMP (Gao et al.), CoBEVT (Xu et al., 2023a), V2X-ViT (Xu et al., 2022a), V2VNet (Wang et al., 2020), FCooper (Chen et al., 2019), AttFuse (Xu et al., 2022b), DiscoNet (Li et al., 2021), and Agent+Feature Selection (AS+FS) method Where2Comm (Hu et al., 2022).

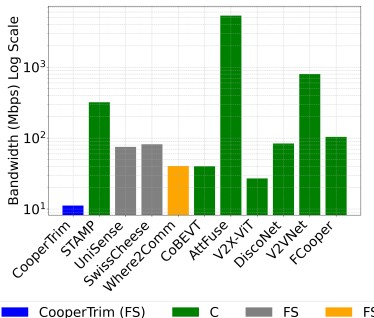

Figure 8: Across-the-board Bandwidth Comparison. C: Compression. FS: Feature Selection. AS: Agent Selection. COOPERTRIM consumes the lowest bandwidth.

Bandwidth comparisons use non-compressed versions (post-selection if applicable) for fairness, reporting original bandwidths from papers or calculating based on our 128x32x32 latent representation size when unspecified. Figure 8 show COOPERTRIM has the lowest network overhead (11.6Mbps) compared to baselines.

**System Overhead.** We compare the system overhead as measured by the processing speed in Frames Per Second (FPS) under static and dynamic settings in the Table 3. The results indicate a modest FPS reduction in COOPERTRIM compared to the baseline, with a decrease of approximately 2 FPS in both configurations. This reduction is due to the additional computational steps in our method. The trade-off is reasonable given the significant bandwidth reduction achieved.

Table 3: System Overhead Comparison. COOPERTRIM has a marginal FPS reduction compared to CoBEVT due to added uncertainty and cross-attention modules for data selection.

| Method | Configuration | FPS | Latency per Frame |
|---|---|---|---|
| CoBEVT | Dynamic | 10.59 | 92.9 ms |
| COOPERTRIM | Dynamic | 8.58 | 94.7 ms |
| CoBEVT | Static | 10.59 | 93.1 ms |
| COOPERTRIM | Static | 8.58 | 119.5 ms |

## 5 CONCLUSION

In this paper, we present COOPERTRIM, an adaptive feature selection framework for cooperative perception that enhances representation learning via temporal uncertainty-driven feature selection for bandwidth-efficient, accurate perception in multi-agent systems. It tackles both relevance (identifying key features) and quantity (optimal sharing based on complexity) assessment. Using $\epsilon$-greedy training, COOPERTRIM balances bandwidth and performance. Evaluated on semantic segmentation and 3D detection, COOPERTRIM achieves up to 80.28% and 72.52% bandwidth reduction, while maintaining accuracy. Compared to baselines, it boosts IoU by 45.54% with 72% less bandwidth, and shows competitive IoU at 1.46% bandwidth usage under 32x compression. COOPERTRIM adapts to dynamics, aiding real-world use, and stays robust to localization errors and communication latency.

ACKNOWLEDGEMENT

This research was sponsored by the OUSD (R&E)/RT&L and was accomplished under Cooperative Agreement Number W911NF-20-2-0267. The views and conclusions contained in this document are those of the authors and should not be interpreted as representing the official policies, either expressed or implied, of the ONR, ARL and OUSD(R&E)/RT&L or the U.S. Government. The U.S. Government is authorized to reproduce and distribute reprints for Government purposes notwithstanding any copyright notation herein.

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

# A APPENDIX

## A.1 LAGRANGE MULTIPLIER ADJUSTMENT

This section explains the dynamic adjustment of Lagrange multiplier for training the loss in Equation 2 The Lagrange multiplier, represented as $\lambda$, acts as a penalty term in the loss function to enforce the constraint on the percentage of selected features. It dynamically adjusts based on training progress (epoch count) to balance the primary perception loss ($L(C(\theta))$) and the bandwidth constraint ($P(C(\theta)) = C_{1.6}$).

- **Initial Phase (epoch $\leq$ ITC)**: For the first few epochs (defined by Initial Tuning Epochs ITC), $\lambda = 0.0$, allowing the model to focus solely on minimizing perception loss without bandwidth constraints for a strong start.
- **Periodic Update (every 10th epoch after initial tuning)**: At every 10th epoch, $\lambda$ scales exponentially with scaling factor SF $= \frac{P(C(\theta))}{100.0}$ and $\lambda = \lambda \cdot 2^{\text{SF}}$, aggressively pushing the model towards the target constraint if far off.
- **Incremental Scaling (other epochs after initial tuning)**: In other epochs, $\lambda$ increases linearly as $\lambda = \lambda \cdot (1 + 0.1 \cdot \lfloor \frac{\text{epoch}-\text{ITC}}{10} \rfloor)$, ensuring a gradual push towards the constraint without abrupt loss changes.

The strategy starts with unconstrained optimization for initial learning, then introduces and intensifies constraint enforcement over time, using periodic strong adjustments for major deviations and steady increments for fine-tuning.

## A.2 MATHEMATICAL PROOF FOR EFFECTIVENESS OF $\epsilon$-GREEDY TRAINING

We provide a detailed proof to demonstrate the effectiveness of the $\epsilon$-greedy training strategy (Section 3.3) by analyzing the bias of the gradient estimator. This proof shows reductions in bias compared to a baseline of using only partial data. We assume perfect transmission, meaning all variability and bias stem from the inherent properties of the data (full or partial) rather than external noise.

### 1. SETUP AND NOTATION

- **Datasets**: Let $D_{\text{full}}$ represent the complete dataset, and $D_{\text{partial}}$ be a subset of $D_{\text{full}}$, representing partial data.
- **True Gradient**: The target gradient to estimate is $\nabla L(D_{\text{full}})$, reflecting the loss over the entire dataset.
- **Transmission Assumption**: Under perfect transmission, computed gradients are unaffected by external noise, so bias or variance arises solely from the data used.
- **$\epsilon$-Greedy Gradient Estimator**: Define the gradient estimator under the $\epsilon$-greedy strategy as $\nabla L_\epsilon$, which is:
    - $\nabla L(D_{\text{full}})$ with probability $\epsilon$
    - $\nabla L(D_{\text{partial}})$ with probability $1 - \epsilon$

### 2. BIAS ANALYSIS OF THE GRADIENT ESTIMATOR

The bias of an estimator is the difference between its expected value and the true value. Here, the true gradient is $\nabla L(D_{\text{full}})$, as full data provides the most accurate representation of the loss landscape.

#### EXPECTED VALUE OF THE ESTIMATOR

The expected value of the $\epsilon$-greedy gradient estimator is:

$$\mathbb{E}[\nabla L_\epsilon] = \epsilon \cdot \mathbb{E}[\nabla L(D_{\text{full}})] + (1 - \epsilon) \cdot \mathbb{E}[\nabla L(D_{\text{partial}})]$$

BIAS CALCULATION

The bias is defined as:

$$\text{Bias}(\nabla L_\epsilon) = \mathbb{E}[\nabla L_\epsilon] - \mathbb{E}[\nabla L(D_{\text{full}})]$$

Substituting the expected value:

$$\text{Bias}(\nabla L_\epsilon) = (\epsilon \cdot \mathbb{E}[\nabla L(D_{\text{full}})] + (1 - \epsilon) \cdot \mathbb{E}[\nabla L(D_{\text{partial}})]) - \mathbb{E}[\nabla L(D_{\text{full}})]$$

Simplifying step-by-step:

$$\begin{aligned}
\text{Bias}(\nabla L_\epsilon) &= \epsilon \cdot \mathbb{E}[\nabla L(D_{\text{full}})] + (1 - \epsilon) \cdot \mathbb{E}[\nabla L(D_{\text{partial}})] - \mathbb{E}[\nabla L(D_{\text{full}})] \\
&= (\epsilon - 1) \cdot \mathbb{E}[\nabla L(D_{\text{full}})] + (1 - \epsilon) \cdot \mathbb{E}[\nabla L(D_{\text{partial}})] \\
&= (1 - \epsilon) \cdot (\mathbb{E}[\nabla L(D_{\text{partial}})] - \mathbb{E}[\nabla L(D_{\text{full}})])
\end{aligned}$$

Thus, the bias expression is:

$$\text{Bias}(\nabla L_\epsilon) = (1 - \epsilon) \cdot (\mathbb{E}[\nabla L(D_{\text{partial}})] - \mathbb{E}[\nabla L(D_{\text{full}})])$$

3. EXPLANATION OF SPECIFIC CASES AND BIAS SCALING

We explore the implications of the bias expression under different scenarios and explain the origin of the bias scaling by $1 - \epsilon$.

CASE 1: UNBIASED PARTIAL DATA GRADIENT

If the partial data gradient is unbiased, i.e., $\mathbb{E}[\nabla L(D_{\text{partial}})] = \mathbb{E}[\nabla L(D_{\text{full}})]$, the difference term is zero:

$$\mathbb{E}[\nabla L(D_{\text{partial}})] - \mathbb{E}[\nabla L(D_{\text{full}})] = 0$$

Thus:

$$\text{Bias}(\nabla L_\epsilon) = (1 - \epsilon) \cdot 0 = 0$$

In this ideal scenario, there is no bias in the $\epsilon$-greedy estimator regardless of $\epsilon$. This occurs when $D_{\text{partial}}$ perfectly represents $D_{\text{full}}$, which is rare in practice due to sampling variability.

CASE 2: BIASED PARTIAL DATA GRADIENT AND ORIGIN OF BIAS SCALING

If there is a systematic difference, i.e., $\mathbb{E}[\nabla L(D_{\text{partial}})] \neq \mathbb{E}[\nabla L(D_{\text{full}})]$, define the inherent bias of using only partial data as:

$$\text{Bias}(\nabla L(D_{\text{partial}})) = \mathbb{E}[\nabla L(D_{\text{partial}})] - \mathbb{E}[\nabla L(D_{\text{full}})]$$

Substituting into the bias expression:

$$\text{Bias}(\nabla L_\epsilon) = (1 - \epsilon) \cdot \text{Bias}(\nabla L(D_{\text{partial}}))$$

Considering the magnitude of bias:

$$|\text{Bias}(\nabla L_\epsilon)| = (1 - \epsilon) \cdot |\text{Bias}(\nabla L(D_{\text{partial}}))|$$

Since $0 \leq \epsilon \leq 1$, the factor $1 - \epsilon < 1$ for any $\epsilon > 0$, implying:

$$|\text{Bias}(\nabla L_\epsilon)| < |\text{Bias}(\nabla L(D_{\text{partial}}))|$$

**Origin of Scaling**: The factor $1 - \epsilon$ comes from the probabilistic weighting in the $\epsilon$-greedy strategy. The expected gradient is a weighted average where the partial data gradient (carrying inherent bias) is weighted by $1 - \epsilon$, and the unbiased full data gradient is weighted by $\epsilon$. Thus, the contribution of the biased gradient is reduced, scaling the bias by $1 - \epsilon$.

4. SUMMARY OF BIAS REDUCTION MECHANISM

- **Scaling Effect**: The bias scales by $1 - \epsilon$ due to the probabilistic blending of full and partial data gradients.

- **Extreme Cases**:
  - When $\epsilon = 0$, only partial data is used, and the bias equals the full inherent bias of the partial data gradient.
  - When $\epsilon > 0$, incorporating full data reduces the weight of the biased partial data gradient to $1 - \epsilon$, scaling down the bias.

- **Bias Comparison**: The inequality $|\text{Bias}(\nabla L_\epsilon)| < |\text{Bias}(\nabla L(D_{\text{partial}}))|$ holds for any $\epsilon > 0$, showing that even a small probability of using full data mitigates bias.

This mechanism highlights the $\epsilon$-greedy approach's advantage in balancing computational efficiency (using partial data) with accuracy (using full data to correct bias).

As shown, the bias of the gradient estimator is reduced by a factor of $(1-\epsilon)$ compared to using only partial data, pulling the expected gradient closer to the true gradient $\mathbb{E}[\nabla L(D_{\text{full}})]$. This improves the accuracy of the optimization direction, leading to better convergence toward the true optimum of $L$. Under perfect transmission, with no external noise, the $\epsilon$-greedy strategy effectively leverages the strengths of full data (lower bias and variance) while maintaining efficiency by using partial data most of the time, achieving a favorable trade-off.

## A.3 PRELIMINARY EXPERIMENTS FOR UNCERTAINTY BASED SELECTION ON SEGMENTATION TASK

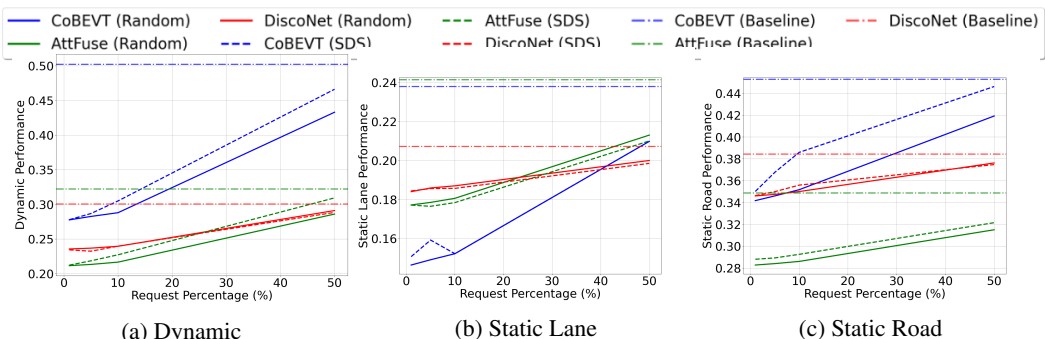

(a) Dynamic      (b) Static Lane      (c) Static Road

Figure 9: Motivation for COOPERTRIM. Comparison of baseline, random channel selection and SD-based selection (SDS) for uncertainty guidance show uncertainty-guided selection often outperforms random selection, though baseline performance remains higher consistently. COOPERTRIM addresses this accuracy gap through its proposed adaptive uncertainty driven selection method.

## A.4 "TRIMMING" COOPERATIVE SEGMENTATION BASELINES - ADDITIONAL

Additional presentation for Section 4 Cooperative Segmentation Baseline experiments in Table 4. COOPERTRIM maintains comparable performance accuracies while reducing bandwidth significantly, achieving an average 80.28% improvement in network overhead over the baselines.

Table 4: Application of COOPERTRIM to Existing Cooperative Segmentation Methods. "Baseline" and "Ours" (COOPERTRIM) show the respective IOU %, while "BW" shows COOPERTRIM bandwidth usage.

| Methods | Dynamic | | | Static Lane | | | Static Road | | | Avg. |
| | Baseline | Ours | BW | Baseline | Ours | BW | Baseline | Ours | BW | BW (Mbps) |
|---|---|---|---|---|---|---|---|---|---|---|
| CoBEVT | 50.23 | 54.03 | 32.04% | 23.79 | 24.45 | 23.77% | 45.28 | 44.38 | 23.77% | 11.16 |
| AttFuse | 32.20 | 30.90 | 24.76% | 24.14 | 23.93 | 17.39% | 34.86 | 36.22 | 17.39% | 8.4 |
| DiscoNet | 30.03 | 30.80 | 10.65% | 20.72 | 22.05 | 9.72% | 38.43 | 40.02 | 9.72% | 4.07 |

A.5    ROBUSTNESS OF THRESHOLD-BASED METHOD IN COOPERTRIM

In COOPERTRIM, the decision to request data is based on a combination of two key factors: the feature representation (which encodes the scene description as perceived by the ego vehicle in the current frame) and the uncertainty estimation (which compares the past comprehensive scene description with the current ego vehicle's scene understanding to detect discrepancies, e.g., objects that existed previously but are no longer present in ego's perception range, or vice versa). This dual mechanism ensures robustness even in subtle change scenarios.

For instance, in a situation where the overall scene appears static (low temporal change), if the ego vehicle's current feature perception changes—due to factors like occlusion or the sudden appearance of a small object such as a pedestrian—our method will detect this deviation. As a result, the system will trigger more data requests to ensure critical information is not missed. Conversely, if both the scene and the ego vehicle's current perception remain largely unchanged, the system will minimize data requests, optimizing bandwidth usage.

To validate this behavior, we analyzed specific frame sequences in our experiments.

- **Frames 1960-1970**: These frames exhibit significant changes in the scene, associated with high data requests (Figure 5). The changes were verified using 4 camera images and ground truth dynamic data segmentation, confirming the system's responsiveness to dynamic scenarios, as shown in Figure 10.

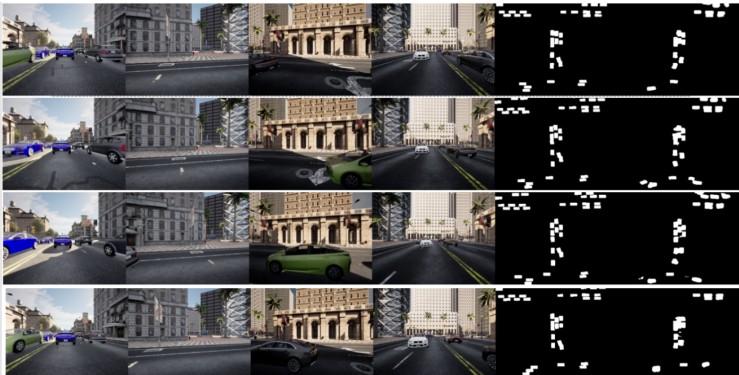

Figure 10: High data requests during significant scene changes in Frames 1960-1970

- **Frames 940-950**: These frames show minimal changes, associated with low data requests (Figure 5), as shown in Figure 10. The lack of significant changes was similarly verified using 4 camera images and ground truth dynamic data segmentation, demonstrating the system's ability to conserve bandwidth in stable conditions.

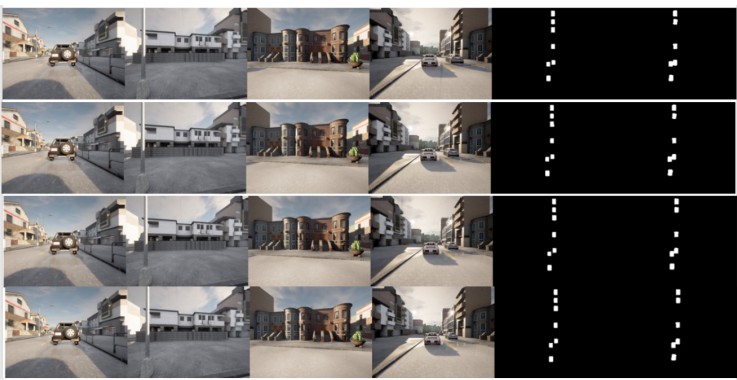

Figure 11: Low data requests during minimal scene changes in Frames 940-950

## A.6 COMPARISON OF COOPERTRIM WITH COMPRESSION-BASED METHODS

We have added comparative results with network-efficient compression-based methods and updated Section 4, where we have compared COOPERTRIM with SwissCheese (Xie et al., 2024) and Where2Comm (Hu et al., 2022), the closest related works in feature selection and communication efficiency. While other feature selection methods like Unisense (Ren et al., 2025) exist, their source code is unavailable for direct comparison. Compression-based approaches, such as those used in CoBEVT (Xu et al., 2023a) and AttFuse (Xu et al., 2022b), are compatible with COOPERTRIM. We show further bandwidth reduction using compression after feature selection. COOPERTRIM significantly outperforms compression-only methods after applying post-selection compression, while maintaining superior perception accuracy. Below are the detailed results for IoU performance and bandwidth usage, with COOPERTRIM achieving a bandwidth usage as low as 1.46% of total data in high-compression settings.

IoU PERFORMANCE COMPARISON AFTER COMPRESSION

| Scenario | COOPERTRIM IoU | CoBEVT IoU | AttFuse IoU |
|---|---|---|---|
| 1x (Dynamic & Static) | 54.03 | 50.23 | 32.20 |
| 1x Lane | 24.45 | 23.79 | 24.14 |
| 1x Road | 44.38 | 45.28 | 34.86 |
| 8x (Dynamic & Static) | 53.99 | 50.19 | 32.21 |
| 8x Lane | 24.52 | 23.78 | 24.13 |
| 8x Road | 44.43 | 45.27 | 34.86 |
| 32x (Dynamic & Static) | 50.32 | 49.63 | 32.19 |
| 32x Lane | 24.62 | 23.77 | 24.14 |
| 32x Road | 45.05 | 45.27 | 34.86 |

BANDWIDTH USAGE COMPARISON AFTER COMPRESSION

| Scenario | COOPERTRIM BW | CoBEVT Compressed BW | AttFuse BW |
|---|---|---|---|
| 1x (Dynamic & Static) | 32.04% | 100% | 100% |
| 1x Lane | 22.77% | 100% | 100% |
| 1x Road | 22.77% | 100% | 100% |
| 8x (Dynamic & Static) | 2.67% | 5.98% | 14.3% |
| 8x Lane | 9.25% | 14.28% | 17.42% |
| 8x Road | 9.25% | 14.28% | 17.42% |
| 32x (Dynamic & Static) | 1.46% | 3.76% | 10.62% |
| 32x Lane | 6.06% | 9.55% | 13.71% |
| 32x Road | 6.06% | 9.55% | 13.71% |

## A.7 TRAINING METHODS COMPARISON - ADDITIONAL

Corresponding to Section 4 Training Methods Comparison, Figure 12 presents the training methods comparative study for training COOPERTRIM, all using the loss function from Section 3.3.

## A.8 NETWORK PERFORMANCE METRICS UNDER REALISTIC CONDITIONS

We evaluated COOPERTRIM's performance by measuring key network-level metrics, including end-to-end latency and packet loss/retransmissions under loss rates of [0–10]%.

### A.8.1 LOSS RATE AND IMPACT

During training, we assumed perfect transmission with no loss. However, during inference, loss rates were simulated randomly between [0, 10]% by applying masks to evaluate robustness. The impact

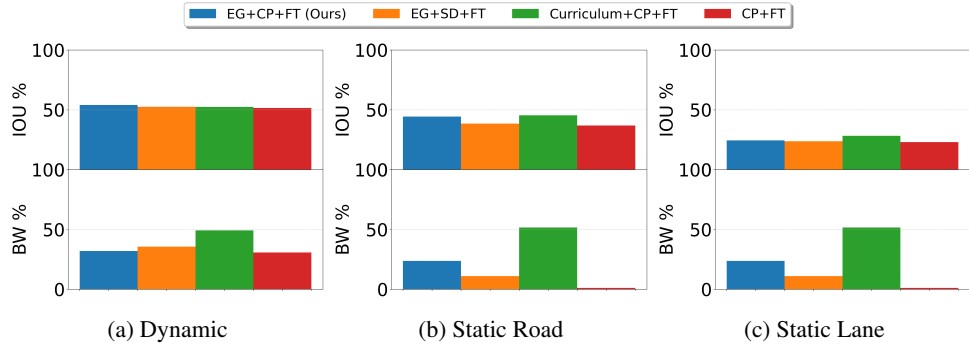

(a) Dynamic      (b) Static Road      (c) Static Lane

Figure 12: Training Methods Comparison. COOPERTRIM balances task performance and network overhead better than other baselines.

Table 5: Impact of Loss Rate on IoU and Bandwidth Metrics. COOPERTRIM maintains stable IoU and BW metrics.

| Loss Rate | Dynamic IoU (%) | Dynamic BW (%) | Static Lane IoU (%) | Static Road IoU (%) | Static BW (%) |
|---|---|---|---|---|---|
| 0% | 54.03 | 32.04 | 24.45 | 44.38 | 23.77 |
| 10% | 53.95 | 32.11 | 24.49 | 44.34 | 23.71 |

on performance metrics such as IoU and bandwidth (BW) under dynamic and static configurations is summarized in Table 5.

Under simulated loss rates of 0–10%, COOPERTRIM maintains stable IoU and bandwidth metrics, with Dynamic IoU being comparable at 10% loss. These results highlight COOPERTRIM's robust fallback behavior under packet loss conditions up to 10%, demonstrating its potential for reliable operation in real-world scenarios with imperfect network conditions.

## A.9 CONFORMAL TEMPORAL UNCERTAINTY

Figure 13 shows the calculation of uncertainty in intermediate representation using conformal prediction inspired quantile gating mechanism explained in Section 3.2.

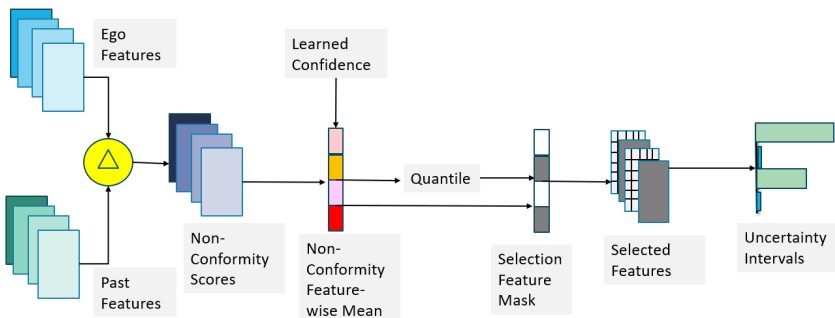

Figure 13: COOPERTRIM estimates conformal temporal uncertainty by comparing the current frame's encoded features with the ego vehicle's fused features from recent frames, highlighting what has changed over time (Non-Conformity Scores). It then uses a learnable quantile-based gating rule to identify high-deviation (more uncertain) features as the most relevant candidates for request.

### A.10   LLM USAGE

In the preparation of this paper, Large Language Models (LLMs) were utilized as a supportive tool for polishing the writing and implementing minor code modifications at specific points. However, the core research ideas, code design, and overall framework are entirely our own. The LLMs played no role in the ideation or conceptualization of the research.

