# OpenReview forum: "COOPERTRIM: Adaptive Data Selection for Uncertainty-Aware Cooperative Perception"
_ICLR.cc/2026/Conference — ICLR 2026 Poster_

### Official Review · Reviewer_Cjxj · 2025-10-29

**Soundness:** 3
**Presentation:** 2
**Contribution:** 3
**Rating:** 6
**Confidence:** 3

**Summary:**

# What the paper does
COOPERTRIM is an adaptive data-selection framework for cooperative perception that uses temporal uncertainty to decide what features to share and how much to share under bandwidth limits.
# Key idea
* Compute a conformal, temporal uncertainty signal by comparing current encoded features (F_t) with the previous fused features (F_{t-1}^{\text{fused}}); uncertainty indicates where collaboration helps most.
* Use an uncertainty-guided attention to score relevance per channel/region and apply adaptive thresholds to (i) select features and (ii) determine sharing quantity frame-by-frame.
* Train with an ε-greedy–inspired regimen to balance exploration/exploitation so the model learns robust selection under bandwidth constraints.

**Strengths:**

1. The temporally driven, uncertainty-aware communication scheme is conceptually clear and well structured: it measures discrepancies between the previous fused representation and the current features, applies conformal quantile thresholding to select candidates, and then uses attention with an adaptive mask cutoff to decide both what to transmit and how much—focusing bandwidth on high-value regions.
2. The $\epsilon$-greedy training schedule provides a practical stabilizer under bandwidth constraints: intermittent full-feature updates interleaved with predominantly masked updates smooth optimization and reduce variance, yielding stronger performance than standard-deviation–only uncertainty baselines and curriculum-style fine-tuning.
3. The method is readily portable: as a drop-in component for cooperative semantic segmentation backbones (e.g., CoBEVT, AttFuse, DiscoNet), it delivers consistent improvements at equal or lower communication budgets.

**Weaknesses:**

1.	Lack of comparison with asynchrony-robust methods (e.g., CoBEVFlow). While the task settings may differ, CoBEVFlow demonstrates that estimating BEV flow and propagating prior features can effectively counter temporal variation; this capability should be considered—either as a baseline or as a complementary design—when claiming advantages in time-varying scenes and realistic, asynchronous communications.
2.	Single-benchmark evaluation. Experiments are confined to OPV2V, which limits external validity. Broader evidence across datasets (e.g., DAIR-V2X, V2X-Sim, OPV2V-Async) and tasks beyond semantic segmentation would strengthen generality claims.
3.	“Conformal” is used primarily as quantile gating rather than as standard conformal prediction with finite-sample coverage guarantees. The paper lacks formal coverage analyses or mismatch bounds, so the terminology risks overstating the method’s theoretical assurances.
4.	Limited system- and communication-layer characterization. Reported metrics focus on bandwidth ratios/Mbps and IoU, with no measurements of end-to-end latency, packet loss/retransmissions, congestion behavior, or the computation/runtime overhead introduced by attention and masking under different hardware budgets. Deployment-level compute-communication trade-offs thus remain underexplored.

**Questions:**

1.	Benchmark against asynchrony-robust methods and/or integrate BEV flow.
Can you compare to CoBEVFlow or prepend a BEV-flow pre-alignment module, reporting IoU–bandwidth trade-offs under controlled time offsets (e.g., ±50/100/200 ms) on OPV2V/OPV2V-Async? Does your two-threshold policy still add gains beyond flow alone?
2.	Report system- and network-level metrics under realistic conditions.
Measure end-to-end latency (encode→select→transmit→align→fuse→decode), packet loss/retransmissions, and congestion behavior across link budgets (e.g., 3/6/12 Mbps) and loss rates (0–10%). Plot IoU–bandwidth–latency curves and characterize degradation/fallback under losses.
3.	Quantify computational overhead and deployment feasibility.
Detail added FLOPs/memory and per-frame latency from attention/masking on embedded automotive hardware (e.g., Jetson/SoC) and desktop GPUs. Compare compute-communication trade-offs against feature-compression/distillation baselines at equal accuracy.
4.	Establish cross-dataset and cross-task generalization.
Evaluate beyond OPV2V (e.g., DAIR-V2X, V2X-Sim, OPV2V-Async) and beyond semantic segmentation (detection/occupancy/tracking). Include fine-tuned and zero-shot transfers, reporting full IoU–bandwidth curves to substantiate external validity.

I would support acceptance provided the authors satisfactorily resolve all identified concerns.

---

> ### Author Response · Authors · 2025-11-21
>
> **Weakness 1 and Question 1:** We sincerely appreciate valuable feedback from the reviewer to evaluate against asynchrony-robust methods and offer a discussion on CoBEVFlow, as well as an analysis of IoU-bandwidth trade-offs under controlled time offsets on the OPV2V dataset. We have updated the detailed results focusing on robustness against synchrony and presented in ***Section 4: Experiments - Sensitivity Analysis linked to Figure 8 and line 432*** of the paper under latency discussion.
>
> **Discussion on CoBEVFlow and Asynchrony Robustness:** CoBEVFlow is an asynchrony-robust collaborative perception system designed to handle temporal misalignments among agents due to communication delays, interruptions, and clock discrepancies in traditional setups where 100% of data from Connected Autonomous Vehicles (CAVs) is sent to the ego vehicle. It leverages Bird’s Eye View (BEV) flow to mitigate information mismatch during multi-agent fusion. In contrast, CooperTrim operates on a partial data request mechanism, selectively acquiring data from CAVs based on uncertainty-driven critical regions, relying predominantly on ego vehicle data. This fundamental difference in data selection strategy offers CooperTrim an inherent resilience to asynchrony, even though it is not explicitly tuned for asynchronous settings during training.
>
> **Simulated Results in Asynchronous Settings:** We have presented the ***results in response to Reviewer Yo96’s feedback on Major Weakness 2 (b)***. Our analysis under controlled time offsets (50ms, 100ms, 200ms) on OPV2V shows that CooperTrim handles asynchrony effectively. Static scenarios (e.g., Lane and Road) show negligible IoU degradation, while dynamic scenarios exhibit a slight drop (from 54.03 to 53.50 at 200ms) due to higher data requests (32.04% in dynamic vs. 23.77% in static). CooperTrim’s focus on limited data (23-32%) reduces dependency on misaligned data, maintaining accuracy with low communication overhead. The ego vehicle’s focus on uncertainty-based critical regions ensures that the majority of the scene is processed with high confidence locally, minimizing dependency on potentially misaligned data from other CAVs. This selective data approach, as demonstrated in ***Figure 7*** of the paper, leads to better feature representation and improved accuracy compared to systems processing entire datasets with varying feature accuracy across CAVs due to visibility and other factors.
>
> **Impact of Two-Threshold Policy Beyond Flow Alone:** The learned two-threshold policy in CooperTrim already adds significant gains by adaptively prioritizing critical data, reducing reliance on external data prone to asynchrony. Unlike flow-alone methods that still process whole shared data, CooperTrim’s policy ensures that only essential data is requested, inherently lessening asynchrony’s impact. Thus the two-threshold policy remains a key differentiator, offering benefits without flow alignment by optimizing data selection for both accuracy and efficiency.
>
> **Weakness 2 and Question 4:** We have expanded the evaluation to 3D detection task and to V2V4REAL dataset in the ***General Response S1***. We have examined closely related works, including CoBEVT, AttFuse, Disconet, Where2Comm, SwissCheese; they do not transfer well in zero-shot. We acknowledge that zero-shot transfer in this particular domain is still underexplored.
>
> **Weakness 3:** We are glad to address the reviewer's nice reminder on the terminology “Conformal” and have updated the clarification in ***Section 3.2 (CooperTrim Design - Conformal Temporal Uncertainty linked to line 186)*** of the paper.
> We acknowledge that our approach uses quantile gating inspired by conformal prediction, not the standard framework with finite-sample coverage guarantees. Our method deviates by (a) learning online frame-wise instead of using a fixed calibration dataset, and (b) estimating uncertainties in components of regression values rather than direct intervals on the entire regression data. We believe that the theoretical guarantees for the conformal prediction inspired uncertainty is an independent direction of work separate from the focus of this paper. Instead we focus on empirical results for BEV feature uncertainty quantification. The term "conformal" reflects conceptual inspiration, not theoretical assurances, and we apologize for any confusion.

---

> ### Author Response · Authors · 2025-11-21
>
> **Weakness 4 and Question 3:** We sincerely appreciate the feedback on computational overhead and deployment feasibility. We have presented the results in this comment.
> To address reviewer’s suggestion on including comparison against compression based methods, we have presented our results in ***General Response S3***  and explained the compute-communication tradeoff in this comment.
>
> **Computational Overhead: FLOPs and Memory**
> The introduction of CooperTrim’s data selection mechanism, which includes uncertainty and cross attention operations, results in slightly increase in overhead by 0.07 GFLOPs and 0.29 million parameters over CoBEVT in both static and dynamic settings, showing minimal computational burden while achieving significant bandwidth savings (41.84% on for detection and 80.28% for segmentation).
>
> | Method        | Configuration | GFLOPs   | Parameters (Millions) |
> |---------------|---------------|----------|-----------------------|
> | CoBEVT        | Static        | 732.29   | 24.14                 |
> | CooperTrim    | Static        | 732.36   | 24.43                 |
> | Difference    | Static        | 0.07     | 0.29                  |
> | CoBEVT        | Dynamic       | 732.25   | 24.14                 |
> | CooperTrim    | Dynamic       | 732.32   | 24.43                 |
> | Difference    | Dynamic       | 0.07     | 0.29                  |
>
> **Per-Frame Latency on Hardware Platforms:** We evaluated per-frame latency on a high-performance desktop GPU, specifically the NVIDIA RTX 6000 Ada Generation.  We have presented the results for system overhead in the General Response S2.
> Evaluations on embedded hardware like NVIDIA Jetson AGX Xavier will validate anticipated higher latencies with CooperTrim's added overhead. These estimates will require validation through testing to meet real-time automotive requirements, which is beyond the scope of this work.
>
> **Compute-Communication Trade-Offs Compared to Baselines:** When comparing compute-communication trade-offs against feature-compression methods like CooperTrim, CoBEVT, and AttFuse, computational overhead and latency are critical factors. CooperTrim introduces a negligible computational increase of just 0.07 GFLOPs and 0.29 million parameters over baselines like CoBEVT in both static and dynamic configurations. However, this results in a minor latency penalty of about 22ms per frame on high-performance hardware such as the NVIDIA RTX 6000 Ada Generation.. This slight overhead demonstrates that CooperTrim’s data selection mechanisms, while effective, add a small cost in processing time.
> Communication analysis compared to compression-based methods has been updated in the ***General Response S3***.
>
> **Weakness 4 and Question3:**
> We sincerely appreciate your feedback from the reviewer on the performance metrics and have conducted experiments to evaluate them. The result for end-to-end latency is put in ***General Response S2***. The network level evaluations are put in ***Appendix A.7 linked to Table 5***. Below is a concise summary:
>
> CooperTrim demonstrates robust performance under realistic conditions with minimal latency overhead compared to the baseline. In dynamic mode, end-to-end latency is only 0.0018 seconds higher, and even in static mode, the overhead remains acceptable at 0.0264 seconds. Under simulated loss rates of 0–10%, CooperTrim maintains stable IoU and bandwidth metrics, with Dynamic IoU being comparable at 10% loss. These results underscore CooperTrim’s reliability in real-world scenarios.
>
> For congestion behavior, we assume consistent network availability, aligning with other close baselines such as CoBEVT, AttFuse, and Disconet. It is challenging to emulate how C-V2X network would behave under congestion with high fidelity. We are indeed in pursuit of that emulation in a parallel effort.

---

### Official Review · Reviewer_ypSd · 2025-10-30

**Soundness:** 2
**Presentation:** 3
**Contribution:** 2
**Rating:** 6
**Confidence:** 4

**Summary:**

The paper proposes an adaptive data selection framework called COOPERTRIM for cooperative perception in autonomous agents. The main idea is to exploit the temporal continuity of the environment to identify relevant features and avoid transmitting redundant or static information. This reduces the communication bandwidth required while maintaining comparable accuracy to existing selection strategies. The proposed framework uses a conformal temporal uncertainty metric to measure feature relevance and a data-driven mechanism to determine the amount of data shared. The evaluation shows significant bandwidth reduction and improved IoU compared to other selection strategies. However, there are some limitations, such as no ablation study on choosing optimal thresholds, only one simulated dataset was used, and the method has not been evaluated on real-world datasets. Additionally, the threshold-based method may not be robust in scenarios where only minor changes occur in the scene. The method is currently only evaluated on segmentation tasks and further evaluation on detection tasks would demonstrate its generalizability. Finally, the impact of the computation cost after introducing this data selection to collaboration perception models needs to be evaluated. Overall, the idea of using temporal uncertainty for data selection is interesting and the theoretical proof is sound, but further research is needed to address the limitations mentioned above.

**Strengths:**

1. The idea of data selection by using temporal uncertainty is interesting.
2. The theoretical proof is sound.
3. The reduction in communication bandwidth consumption in segmentation tasks is obvious.

**Weaknesses:**

1. No ablation studies on how to choose the optimal thresholds.
2. Only one simulated dataset is used; No real-world dataset is evaluated.
3. What is the mathematical expression of the distance function? What is the deep reason for using this distance function?
4. How robust is the threshold-based method? For example, in a certain scenario, maybe the overall scene doesn't change much, only a small object (e.g., a new pedestrian emerges), probably leading to a small temporal uncertainty, and how will the system take actions to this?
5. Although the method is advantageous for segmentation tasks, it should be better evaluated on detection tasks as well to demonstrate its generalizability.
6. Apart from the bandwidth consumption, one important factor is how the computation costs change after introducing this data selection to the collaboration perception models. What is the processing speed (evaluated by FPS)? Can this method be used in a real-time driving system (Typically more than 100 FPS)?

**Questions:**

Please refer to the weaknesses.

---

> ### Author Response · Authors · 2025-11-21
>
> **Weakness 4:** We thank reviewer's insightful feedback regarding the robustness of the threshold-based method in CooperTrim. We have updated ***Appendix A.6 linked to Figures 11 and 12 and line 869*** with a detailed analysis and visual intuition, including plots of data selection behavior in Frames 940-950 and 1960-1970, to demonstrate the system's responsiveness and efficiency. Our analysis in CooperTrim shows that the system effectively detects subtle scene changes, triggering data requests for critical updates, while conserving bandwidth in stable conditions.
>
> **Weakness 5:** We appreciate the feedback from the reviewer and have addressed Weakness 5 by presenting results for system overhead in the ***General Response S2***.
>
> **Weakness 1:** We thank reviewer for the question on ablation study of optimal thresholds. We would like to clarify that CooperTrim employs a learned data-driven threshold mechanism that is trained end-to-end. This approach dynamically adapts to scene complexity.
>
> **Weakness 2:** We thank the reviewer for the question regarding real-world dataset evaluation. To address this, we have expanded our evaluation to V2V4Real, a dataset that includes data collected from real-world scenarios. We present the results of our evaluation on V2V4Real in ***General Response S1***. These results demonstrate the applicability of CooperTrim in real-world V2V settings, achieving comparable performance to the baseline with significant bandwidth reduction.
>
> **Weakness 3:** We are glad to address the query regarding the distance function used in Conformal Temporal Uncertainty and have provided the mathematical expression and reasoning for the distance function used in CooperTrim, and have updated it in ***Section 3.2 (CooperTrim Design - Conformal Temporal Uncertainty linked to line 184)*** of the paper.
> The distance function employed in CooperTrim is the L1 distance. Mathematically, for two values \\( F_t \\) (ego feature at frame t) and \\( F_{t-1}^{\\text{fused}} \\) (fused feature of frame t-1), it is expressed as: \\( L1 \\text{ Distance} = |F_t - F_{t-1}^{\\text{fused}}| \\). L1 distance effectively captures the deviation between past comprehensive scene understanding and the current single ego vehicle’s scene understanding. This deviation helps estimate uncertainty by identifying discrepancies—i.e., whether an object that existed in the past is no longer present, or vice versa. The simplicity and interpretability of L1 distance make it suitable for quantifying such differences in a straightforward manner, aligning with our goal of real-time uncertainty estimation.
>
> **Weakness 6:** We thank reviewer for the question. We have addressed it in ***General Response S2***.

---

### Official Review · Reviewer_Yo96 · 2025-11-01

**Soundness:** 3
**Presentation:** 3
**Contribution:** 2
**Rating:** 4
**Confidence:** 4

**Summary:**

This paper presents CooperTrim, an adaptive, uncertainty-aware feature selection framework for cooperative perception. The framework leverages conformal prediction to estimate temporal uncertainty for feature relevance and employs a data-driven adaptive mechanism to select an appropriate quantity of shared features based on environmental complexity. CooperTrim is plugged in and extensively evaluated on semantic segmentation using several co-perception methods on the OPV2V dataset, demonstrating significant reductions in bandwidth usage without sacrificing accuracy.

**Strengths:**

**Compelling Motivation and Scope:** The paper focuses on the bandwidth-accuracy trade-offs in cooperative perception, arguing for temporally and contextually adaptive feature selection that is not covered by static or threshold-based approaches.

**Effectiveness of Sub-modules:** CooperTrim's integration of conformal temporal uncertainty estimation with a cross-attention-based selection mechanism is well described, addressing both feature relevance and adaptivity. Detailed elaboration on training strategies provides hints on reproduction.

**Concrete Adaptivity Insights:** The claim of scene adaptivity is validated by the qualitative results given in Fig. 4.

**Weaknesses:**

**Major Weaknesses:**
1. Although more components are incorporated, the proposed method remains a threshold masking mechanism. The adaptivity claim needs more quantitative validation. For example, in Fig. 4 (left), a convincing result would be to show that the IoU curve is relatively stable, or at least more stable than the BW curve ("complexity" curve). The current result cannot prove that the adaptivity benefits the final results.
2. Robustness against localization error and latency is not discussed.

**Minor Weaknesses:**
1. The method is only tested on OPV2V, which is a relatively simple simulated dataset.
2. More network-efficient baselines are expected to be included. Presenting results for the object detection task can also help enhance comparisons with previous methods.

**Questions:**

1. Can the method adjust the bandwidth requirement during operation? Or a new model (a newly learned threshold generator) is needed to cope with a new network condition?
2. There will be a potential delayed response to a new traffic pattern as history information is used to generate the threshold. Are there any results on the influence of the temporal window size?
3. How well would CooperTrim's adaptivity generalize to other perception tasks (detection, tracking, etc.)? In those tasks, the ROI is much sparser.

See the weaknesses section for more details.

---

> ### Author Response · Authors · 2025-11-21
>
> **Major Weakness 2:** We appreciate the constructive feedback and have conducted additional experiments evaluating the robustness against localization error and latency. The following results and analysis have been put in ***Section (4) Experiments - Sensitivity Analysis  linked to Figures 6 and 8 and line 418*** of the paper.
>
> **(a) Robustness Against Localization Error:** CooperTrim shows reasonable robustness against small localization errors (up to  ±20cm), with comparable IoU at larger error of  ±1m. We observe that bandwidth remains largely unchanged, indicating stability in communication efficiency. Below is a table summarizing the performance under localization errors [0cm, ±20cm, ±1m]:
>
> | Localization Error | Dynamic IoU (%) | Dynamic BW (%) | Static Lane IoU (%) | Static Road IoU (%) | Static BW (%) |
> |--------------------|-----------------|----------------|---------------------|---------------------|---------------|
> | 0cm                | 54.03           | 32.04          | 24.45               | 44.38               | 23.77         |
> | ±20cm              | 53.69           | 32.05          | 24.19               | 44.16               | 23.77         |
> | ±1m                | 48.54           | 32.06          | 21.85               | 42.66               | 23.74         |
>
>
> **(b)Robustness Against Latency:** To address the reviewer's comment, we evaluate CooperTrim under various latency conditions (0ms to 200ms), with results added to the same section. Below is a table summarizing the performance under latency:
>
> | Latency (ms) | Dynamic IoU (%) | Static Lane IoU (%) | Static Road IoU (%) |
> |--------------|-----------------|---------------------|---------------------|
> | 0            | 54.03           | 24.45               | 44.38               |
> | 50           | 54.03           | 24.45               | 44.38               |
> | 100          | 53.88           | 24.44               | 44.38               |
> | 200          | 53.50           | 24.38               | 44.29               |
>
>
>
> CooperTrim exhibits strong robustness against latency up to 50ms, with no performance loss. At 100ms and 200ms, the IoU degradation remains minimal, especially for Static Road IoU, which shows remarkable stability. This suggests that CooperTrim's adaptive mechanisms, using its two-threshold policy, effectively mitigate the impact of latency up to moderate levels. Furthermore, CooperTrim's selective data request strategy, focusing on uncertainty-driven critical regions, ensures that the majority of the scene is processed locally with high confidence, reducing dependency on potentially misaligned data from other Connected Autonomous Vehicles (CAVs). By requesting significantly less data overall (23-32% compared to baseline as shown in Appendix Table 4), the impact of asynchrony is substantially minimized, making it particularly effective in real-world scenarios.
>
>
>
> **Major Weakness 1**: We appreciate the feedback from Reviewer Yo96 on the adaptivity claim and have updated ***Section (4) Experiments - Environment Adaptation linked to Figure 5 and line 359*** with detailed quantitative validation, demonstrating CooperTrim’s effectiveness over baselines. While achieving a completely flat IoU curve is challenging due to inherent scene complexity variations that affect all methods, including the baselines with infinite bandwidth assumptions, our results show significant performance gains with reduced bandwidth usage.
> Specifically, we added baseline IoU in ***Figure 5***. In critical frame ranges such as 1000-1800 (Dynamic), 1000-1500 (Lane), and 1200-1400 (Road), where baselines consistently underperform, CooperTrim achieves much higher IoU, illustrating the effectiveness of its adaptive threshold masking mechanism in prioritizing key features and managing scene complexity.

---

> ### Author Response · Authors · 2025-11-21
>
> **Minor Weakness 1:** We appreciate the feedback from the reviewer and have addressed Minor Weakness 1 by presenting results for the object detection task in the ***General Response S1***.
>
> **Minor Weakness 2:** The 2 questions in MInor Weakness 2 has been addressed in the ***General Response S3 and S1 respectively***.
>
> **Question 1:** We thank the reviewer for this fascinating question. While our focus was on reducing network bandwidth adaptive to scene complexity, we would be thrilled to explore adaptivity to network conditions. We acknowledge that integrating cross-stack adaptivity—combining perception-based bandwidth reduction with real-time network condition monitoring—is a non-trivial challenge. Currently, our method does not support adjusting bandwidth requirements. Though most open-source baselines, such as CoBEVT, AttFuse, and Disconet assume a stable network, we very much agree with the reviewer on the sentiment of more network-aware adaptation.
>
> **Question 2:** We appreciate the reviewer's concern regarding potential delayed responses to new traffic patterns due to the use of historical information for threshold generation. We would like to clarify that our method does not experience a delayed response, as it operates on a per-frame basis, using immediate data from the previous frame to make decisions. This ensures real-time adaptability to changing conditions without reliance on a historical temporal window.
> Nevertheless, investigating the effect of varying temporal window sizes on performance and adaptability is an interesting direction, which we plan to explore to further enhance our method.
>
> **Question 3:** We have addressed this in the ***General Response S1***.

---

> ### Comment · Reviewer_Yo96 · 2025-11-24
>
> Thank the authors for the clarification, and most of my concerns have been resolved. The updated empirical results on adaptivity and robustness are helpful. The scenario adaptivity comparison between the baseline (Which baseline does the result come from?) and CooperTrim clearly shows that the IoU curve is "relatively flattened", which is more convincing to me than the absolute performance improvement. This better distinguishes the proposed CooperTrim from the existing threshold-based method, and I will raise my rating after the discussion period.

---

> > ### Author Response · Authors · 2025-11-25
> >
> > We appreciate the reviewer’s feedback on our revisions and for highlighting the oversight regarding the omission of the baseline name. We have used CoBEVT as our baseline. Accordingly, in the camera-ready version, we will update ***Section (4) Experiments - Environment Adaptation linked to Figure 5, line 367*** and explicitly mention "CoBEVT" as the baseline. Additionally, we will include the baseline information in the description of ***Figure 5*** for further clarity.

---

> > > ### Author Response · Authors · 2025-11-25
> > >
> > > We have updated the details of baseline as CoBEVT in the paper ***Section (4) Experiments - Environment Adaptation linked to Figure 5, line 367*** and in the caption of ***Figure 5***.

---

### Author Response · Authors · 2025-11-21

**General Response I:**

We thank all reviewers for the constructive feedback. We are encouraged by the positive assessments. We are addressing concerns shared across reviewers (S1-S3) in this general response, and specific concerns with more details in individual responses.

**S1: Evaluation of CooperTrim on a task other than segmentation and on a dataset other than OPV2V. (Reviewers- All)**

**Response:** We extend our evaluation of CooperTrim to 3D detection task and V2V4REAL dataset, to demonstrate the robustness and generalizability of our approach. We also clarify why semantic segmentation task evaluation was constrained to specific datasets. The following results have been attached below and added to the paper: ***Section (4) Experiments- “Trimming” Existing Cooperative Perception Baselines linked to Figure 2(d) and line 301***.

**Limitations on Semantic Segmentation Across Datasets:** Beyond OPV2V dataset, unfortunately, several other prominent datasets in the vehicle-to-vehicle (V2V) and vehicle-to-everything (V2X) domains, such as DAIR-V2X, V2XSeq, and V2V4Real, do not include semantic segmentation groundtruth. Specifically:
DAIR-V2X focuses primarily on detection tasks and lacks semantic segmentation annotations, making it unsuitable for evaluating segmentation performance.
V2XSeq supports tracking and detection tasks but similarly does not provide semantic segmentation GT data.
V2V4Real includes 3D vector maps and supports detection tasks, but it lacks semantic segmentation annotations.
Additionally, prior work such as CoBEVT has demonstrated segmentation performance on datasets like nuScenes and OPV2V. NuScenes is a single agent dataset and hence not considered for our work.

**Extention to 3D Detection Task on both OPV2V and V2V4REAL:** To address concerns on CooperTrim compatibilityty and dataset diversity, we extended our evaluation to 3D detection task on both OPV2V and V2V4Real as shown below.  We compare CooperTrim against the baseline CoBEVT across two IoU thresholds (0.5 and 0.7).For now, we provide evaluation on a subset of data for detection. We expect the trends to be the same on the whole dataset and produce the final results by December 2 (***updated in General Response S1(b)***). We also report the bandwidth reduction achieved by each method.

3D Detection Results
| Dataset   | CoBEVT       | CooperTrim   | CoBEVT      | CooperTrim | Bandwidth Use |
|-----------|--------------|--------------|-------------|------------|---------------|
|           | AP@IoU 0.5   | AP@IoU 0.5   | AP@IoU 0.7  | AP@IoU 0.7 | wrt CoBEVT    |
|-----------|--------------|--------------|-------------|------------|---------------|
| OPV2V     | 0.88         | 0.80         | 0.78        | 0.69       | 56.84%        |
| V2V4Real  | 0.61         | 0.53         | 0.42        | 0.31       | 26.83%        |



Consistent with the trends shown by the semantic segmentation task on OPV2V, the detection results show that CooperTrim achieves performance comparable to the baseline, while offering a significant average bandwidth reduction of 42.36%. These findings underscore the practical applicability of our method in real-world V2V scenarios in V2V4Real dataset, balancing performance and efficiency effectively in both - lidar based detection and camera based segmentation tasks. We hope this multi-sensor, multi-task and multi-dataset evaluation strengthens the validity of our approach and addresses the concerns.

---

> ### Author Response · Authors · 2025-11-25
>
> **General Response S1(b):**
>
> We have updated the paper with the results on the ***whole datasets*** for OPV2V and V2V4Real for the 3D detection task in ***Section (4) Experiments - “Trimming” Existing Cooperative Perception Baselines linked to Figure 2(d) and line 301***. As promised, we are sharing the final results on the complete datasets for the 3D detection task. The table below presents the performance metrics for CoBEVT and CooperTrim across both datasets at IoU thresholds of 0.5 and 0.7, along with the bandwidth reduction achieved.
>
> **3D Detection Results on Whole Datasets**
>
> | Dataset      | CoBEVT (AP@IoU 0.5) | CooperTrim (AP@IoU 0.5) | CoBEVT (AP@IoU 0.7) | CooperTrim (AP@IoU 0.7) | Bandwidth Use wrt CoBEVT |
> |--------------|----------------------|-------------------------|----------------------|-------------------------|--------------------------|
> | OPV2V        | 0.94                | 0.90                   | 0.90                | 0.79                   | 24.09%                  |
> | V2V4Real     | 0.72                | 0.60                   | 0.38                | 0.32                   | 30.86%                  |
>
> These results follow the same trend observed in the subset evaluation, with CooperTrim achieving performance comparable to the baseline CoBEVT while significantly reducing bandwidth usage by an average of 27.48%. Hence, we can see CooperTrim effectively balances performance and efficiency in real-world V2V scenarios across multiple tasks and datasets.

---

### Author Response · Authors · 2025-11-21

**General Response II:**

**S2: Evaluation of System Overhead of CooperTrim compared to Baselines. (Reviewers- ypSd Weakness 6 and Cjxj Question 2 and 3 )**

**Response:** We thank Reviewers for the kind reminder on the computation costs and processing speed, for which we conduct the following evaluation and have updated it in ***Section 4.1 (System Overhead) linked to Table 3 and line 441*** of the paper.
We present the processing speed evaluated in Frames Per Second (FPS) for both the baseline CoBEVT and CooperTrim under static and dynamic settings. End-to-end latency per frame was measured across the entire pipeline (encode → select → transmit → align → fuse → decode) for both CooperTrim and the baseline CoBEVT under dynamic and static configurations. CooperTrim adds a minor latency of ~14ms per frame, reducing FPS from 10.69 to 8.67 in dynamic settings.

| Method       | Configuration | FPS   | Time(ms) per Frame (end-to-end) |
|--------------|---------------|-------|---------------------------------|
| CoBEVT       | Dynamic       | 10.69 | 92.9                            |
| CooperTrim   | Dynamic       | 8.67  | 94.7                            |
| CoBEVT       | Static        | 10.59 | 93.1                            |
| CooperTrim   | Static        | 8.58  | 119.5                           |

This indicates a modest reduction in processing speed due to the added computational steps of our method. However, this trade-off is justified by the significant bandwidth reduction achieved, which is the primary focus of our work. Regarding the suitability for real-time driving systems, CooperTrim is focused on perception rather than control frame rate. For reference, Waymo Open dataset which is collected with real-world vehicles has a frame rate of 10 FPS (e.g. LiDAR frame rate).

**S3: Comparison of CooperTrim with Compression-based Methods (Reviewers- Yo96 Minor Weakness 2 and Cjxj Question 3 )**

We have added comparative results with network-efficient compression-based methods and updated ***Section (4) Experiments - Compression-based Method Comparison  linked to Figure 4 and line 317***, where we had compared CooperTrim with SwissCheese and Where2Comm, the closest related works in feature selection and communication efficiency. While other feature selection methods like Unisense exist, their source code is unavailable for direct comparison. Compression-based approaches, such as those used in CoBEVT and AttFuse, are compatible with CooperTrim. We show further bandwidth reduction using compression after feature selection.
CooperTrim significantly outperforms compression-only methods after applying post-selection compression, while maintaining superior perception accuracy. Below are the detailed results for IoU performance and bandwidth usage, with CooperTrim achieving a bandwidth usage as low as 1.46% of total data in high-compression settings.
Table 1: IoU Performance Comparison after Compression
| Scenario   | CooperTrim IoU | CoBEVT IoU | AttFuse IoU |
|------------|----------------|------------|-------------|
| 1x Dyn     | 54.03          | 50.23      | 32.20       |
| 1x Lane    | 24.45          | 23.79      | 24.14       |
| 1x Road    | 44.38          | 45.28      | 34.86       |
| 8x Dyn     | 53.99          | 50.19      | 32.21       |
| 8x Lane    | 24.52          | 23.78      | 24.13       |
| 8x Road    | 44.43          | 45.27      | 34.86       |
| 32x Dyn    | 50.32          | 49.63      | 32.19       |
| 32x Lane   | 24.62          | 23.77      | 24.14       |
| 32x Road   | 45.05          | 45.27      | 34.86       |


Table 2: Bandwidth Usage Comparison after Compression
| Scenario   | CooperTrim BW | CoBEVT Compressed BW | AttFuse BW |
|------------|---------------|----------------------|------------|
| 1x Dyn     | 32.04%        | 100%                 | 100%       |
| 1x Lane    | 22.77%        | 100%                 | 100%       |
| 1x Road    | 22.77%        | 100%                 | 100%       |
| 8x Dyn     | 2.67%         | 5.98%                | 14.3%      |
| 8x Lane    | 9.25%         | 14.28%               | 17.42%     |
| 8x Road    | 9.25%         | 14.28%               | 17.42%     |
| 32x Dyn    | 1.46%         | 3.76%                | 10.62%     |
| 32x Lane   | 6.06%         | 9.55%                | 13.71%     |
| 32x Road   | 6.06%         | 9.55%                | 13.71%     |

---

### Author Response · Authors · 2025-12-01

**Summary of Modifications and Experiments During Rebuttal**:

We sincerely thank the reviewers for their insightful feedback during the review process for CooperTrim. Below, we summarize (a) the key modifications and experiments conducted during the rebuttal phase in response to the reviewers' comments, (b) the paper updates, and (c) the clarifications provided. We also (d) acknowledge Reviewer Yo96's positive feedback and decision to raise their rating.

***(a) Summary of Key Modifications and Experiments:***

- (1) Extended CooperTrim evaluation to 3D detection tasks on OPV2V and V2V4Real datasets, showing comparable performance to baseline CoBEVT with significant bandwidth reduction (average 27.48% bandwidth compared to CoBEVT).
- (2) Evaluated system overhead, showing minor latency increase (~14ms per frame) compared to CoBEVT, justified by bandwidth savings.
- (3) Compared with compression-based methods, demonstrating CooperTrim’s superiority (bandwidth usage as low as 1.46%) with better accuracy.
- (4) Conducted sensitivity analysis, confirming robustness to localization errors (up to ±1m) and latency (comparable performance upto 200ms).
- (5) Validated adaptivity with higher IoU in critical frame ranges compared to CoBEVT.
- (6) Detailed computational overhead (0.07 GFLOPs, 0.29M parameters increase) and per-frame latency, balancing compute-communication trade-offs.

***(b) Paper Updates:***

- (1) **Section 4 - “Trimming” Existing Cooperative Perception Baselines (Figure 2(d), line 301):** Showed generalizability across tasks and datasets with bandwidth reduction (bandwidth 27.48%).
- (2) **Section 4 - Compression-based Method Comparison (Figure 4, line 317):** Demonstrated superiority over compression methods (bandwidth 1.46%).
- (3) **Section 4 - Environment Adaptation (Figure 5, line 359):** Validated adaptivity with higher IoU in critical frames.
- (4) **Section 4 - Sensitivity Analysis (Figures 6 & 8, line 418):** Confirmed robustness to errors and latency.
- (5) **Section 4.1 - System Overhead (Section 4.1, Table 3, line 441):** Detailed system overhead.
- (6) **Section 3.2 (CooperTrim Design - Conformal Temporal Uncertainty, line 184 & line 186):** Added L1 distance expression and explained "conformal" terminology; clarified conceptual inspiration.
- (7) **Appendices A.6 (Figures 11 & 12, line 869):** Showed learned threshold based adaptivity's robustness to subtle changes.
- (8) **Appendices A.7 (Table 5, line 941):** Demonstrated stable IoU under loss rates (0–10%).

***(c) Clarifications Provided:***

**Delayed Response to Traffic Patterns (Reviewer Yo96 Question 2):** Clarified CooperTrim operates per-frame with immediate data, ensuring real-time adaptability. Located in response to Reviewer Yo96 (Question 2).

***(d) Acknowledgment of Reviewer Yo96's Feedback:*** We are grateful to Reviewer Yo96 for their constructive feedback and for acknowledging the resolution of most concerns. We are pleased to note that, based on updated results on adaptivity, robustness and clarifications, **Reviewer Yo96 mentioned, *on Nov 24th,* to raise their rating after the discussion period, and had raised the score from 4 to 6, *on Nov 27th*.**

---

### Meta-Review · Area_Chair_HofK · 2026-01-07

**Summary:**

Reviewer concerns primarily focused on limited evaluation scope, generalizability beyond a single simulated dataset, robustness to localization error, latency and asynchrony, computational overhead and real-time feasibility, and deployment-level communication behavior. These concerns were largely addressed through extended experiments on additional tasks (3D detection) and real-world datasets (V2V4Real), added comparisons with stronger baselines, and new analyses of system overhead, robustness to localization error, latency, and packet loss under controlled settings. However, CooperTrim does introduce additional latency, and some deployment-level concerns, such as network congestion behavior and evaluation on embedded hardware is not fully addressed.

**Reviewer Concerns:**

Reviewer Yo96:

Addressed: The authors addressed reviewer concerns by extending CooperTrim to additional tasks and datasets, adding comparisons with stronger baselines, and providing analyses of system overhead, robustness to localization error and latency, and adaptivity to scene complexity. The added results demonstrate that CooperTrim achieves substantial bandwidth reduction with comparable task performance and modest computational overhead. These revisions resolved the main concerns raised by reviewers and increased confidence in the method’s effectiveness and generalizability.

Reviewer ypSd:

Addressed: Concerns regarding limited evaluation scope were addressed by extending experiments to real-world datasets (V2V4Real) and additional tasks (3D detection), demonstrating generalizability beyond simulated segmentation settings. The robustness of the threshold-based method was examined through additional analyses and visualizations showing responsiveness to subtle scene changes. Clarifications were also provided on the mathematical form and motivation of the distance function used in conformal temporal uncertainty.


Partially addressed: Questions about computational cost and real-time feasibility were addressed through added system-overhead analysis reporting FPS and end-to-end latency.  But the results show that CooperTrim introduces additional latency.

Reviewer Xjxj

Addressed: Concerns regarding limited evaluation scope were addressed by extending experiments to additional tasks (3D detection) and real-world datasets (V2V4Real). Robustness to temporal asynchrony was evaluated under controlled latency settings, and the authors clarified conceptual differences with flow-based methods such as CoBEVFlow. Computational overhead was quantified through analyses of FLOPs, parameters, and per-frame latency. Robustness to packet loss was evaluated under simulated loss rates.
Partially addressed: Network-level behaviors such as congestion dynamics, retransmissions, and deployment on embedded automotive hardware were not evaluated.

**Reviewer Scores:**

Reviewer Yo96 would possibly raise their score from 4 to 6, as also mentioned by the reviewer.
Reviewer ypSd would possibly maintain their score of 6.
Reviewer Xjxj would possibly maintain their score of 6.

---

### Decision · Program_Chairs · 2026-01-26

Accept (Poster)